# An alpaca nanobody neutralizes SARS-CoV-2 by blocking receptor interaction

Leo Hanke [1], Laura Vidakovics Perez [1], Daniel J. Sheward[1,2], Hrishikesh Das[3], Tim Schulte[4], Ainhoa Moliner-Morro[1], Martin Corcoran[1], Adnane Achour[4], Gunilla B. Karlsson Hedestam[1], B. Martin Hällberg [3,5✉], Ben Murrell [1,6✉] & Gerald M. McInerney [1,6✉]

SARS-CoV-2 enters host cells through an interaction between the spike glycoprotein and the angiotensin converting enzyme 2 (ACE2) receptor. Directly preventing this interaction presents an attractive possibility for suppressing SARS-CoV-2 replication. Here, we report the isolation and characterization of an alpaca-derived single domain antibody fragment, Ty1, that specifically targets the receptor binding domain (RBD) of the SARS-CoV-2 spike, directly preventing ACE2 engagement. Ty1 binds the RBD with high affinity, occluding ACE2. A cryo-electron microscopy structure of the bound complex at 2.9 Å resolution reveals that Ty1 binds to an epitope on the RBD accessible in both the 'up' and 'down' conformations, sterically hindering RBD-ACE2 binding. While fusion to an Fc domain renders Ty1 extremely potent, Ty1 neutralizes SARS-CoV-2 spike pseudovirus as a 12.8 kDa nanobody, which can be expressed in high quantities in bacteria, presenting opportunities for manufacturing at scale. Ty1 is therefore an excellent candidate as an intervention against COVID-19.

[1] Department of Microbiology, Tumor and Cell Biology, Karolinska Institutet, Stockholm, Sweden. [2] Division of Virology, Institute of Infectious Diseases and Molecular Medicine, Faculty of Health Sciences, University of Cape Town, Cape Town, South Africa. [3] Department of Cell and Molecular Biology, Karolinska Institutet, Stockholm, Sweden. [4] Science for Life Laboratory, Department of Medicine Solna, Karolinska Institutet, and Division of Infectious Diseases, Karolinska University Hospital, Solna, Stockholm, Sweden. [5] Karolinska Institutet VR-RÅC, Centre for Structural Systems Biology, Notkestraße 85, 22607 Hamburg, Germany. [6] These authors contributed equally: Ben Murrell, Gerald M. McInerney. ✉email: martin.hallberg@ki.se; benjamin.murrell@ki.se; gerald.mcinerney@ki.se

SARS-CoV-2 was first identified as the etiologic agent of the novel pneumonia COVID-19, in late 2019. In the comparatively short time since then, it has achieved pandemic status, causing more than 13.3 million cases, leading to more than 600,000 deaths. Accordingly, the WHO declared the pandemic to be a public health emergency of international concern. A safe and effective vaccine is urgently needed, but requires time to develop. In the meantime, and indeed also in the post-vaccine era, highly specific and potent antiviral interventions are needed. Many generic or repurposed candidates are in trials, but so far results have been unremarkable. Since the virus is newly emerged, specifically designed drugs have not yet reached late phase trials. When available, specific antiviral drugs or antibody therapies will be used to protect individuals at risk and their widespread use will allow immunologically naïve populations to exit lockdowns more safely.

The virus is closely related to SARS-CoV-1, both being members of the lineage 2 betacoronaviruses. Cell entry of both viruses is achieved by first binding to the cell surface expressed receptor angiotensin-converting enzyme 2 (ACE2), followed by conformational changes in the viral spike glycoprotein trimer and subsequent membrane fusion. The affinity of SARS-CoV-2 receptor-binding domain (RBD) for ACE2 is considerably higher than that for SARS-CoV-1[1,2], supporting efficient cell entry and likely contributing to pathogenesis. The RBD is a globular domain situated on the distal surface of the spike protein. Two conformations have been observed in the stabilized trimer. Specifically, one conformation where one RBD is ACE2-accessible while the other two are not, and one conformation where all three RBDs are down, i.e. receptor inaccessible[2,3]. As the receptor-engaging part of the spike, the RBD is an attractive target for coronavirus neutralization, and a number of conventional neutralizing monoclonal antibodies that target the RBD and block receptor binding have already been isolated from convalescent patients[4–6].

Camelid-derived single domain antibody fragments, also called VHHs or nanobodies, offer several advantages over conventional antibodies as candidates for specific therapies. Despite being approximately one-tenth of the size of a conventional antibody, they retain specificity and affinity similar to conventional antibodies, while being far easier to clone, express, and manipulate. They are readily expressed in bacteria in large quantities and show high thermal stability and solubility, making them easily scalable and cost effective. Their modularity means that they can be oligomerized to increase avidity or to increase serum half-life[7]. Critical to their use as antivirals in humans, they can easily be humanized with existing protocols[8]. Importantly, they have proven to be highly potent inhibitors of viral infections in vivo; particularly respiratory infections[9,10].

Here, we describe the isolation, evaluation, and molecular determination of an alpaca-derived nanobody, Ty1, directed to the RBD of the SARS-CoV-2 spike glycoprotein. We demonstrate that the monomeric 12.8 kDa Ty1 molecule potently neutralizes SARS-CoV-2 spike pseudovirus. The nanobody binds with high affinity to the RBD in a manner that occludes ACE2 interaction. We have also determined the mechanism of neutralization to be due to direct interference with RBD binding to ACE2. Altogether, these results highlight the great potential of Ty1 as a SARS-CoV-2 antiviral agent.

## Results

**Isolation of a SARS-CoV-2 neutralizing nanobody.** We immunized one alpaca with SARS-CoV-2 S1-Fc and RBD on a 60-day immunization schedule. We generated a phage display library and performed two consecutive rounds of phage display, followed by an ELISA-based binding screen (Fig. 1a). We isolated one nanobody, Ty1, that binds specifically to the RBD of the SARS-CoV-2 spike glycoprotein. In parallel we performed next generation sequencing (NGS) on the baseline and post-enrichment libraries, and quantified variant frequency before and after each enrichment step. Ty1 exhibited the greatest fold-change in frequency among all nanobody variants, increasing over 10,000-fold from baseline to after the second enrichment round (Fig.1b). We report the amino acid sequence of Ty1 in Fig. 1c.

To determine whether Ty1 neutralized SARS-CoV-2 we employed an in vitro neutralization assay using lentiviral particles pseudotyped with the SARS-CoV-2 spike protein. Ty1 neutralized SARS-CoV-2 pseudotyped viruses at an $IC_{50}$ of 0.77 µg/ml (54 nM) (Fig. 2a). No neutralization of a lentivirus pseudotyped with VSV-G by Ty1 was evident, and a control nanobody produced and purified in the same way, but specific for influenza A virus nucleoprotein[11], showed no evidence of neutralization of SARS-CoV-2 pseudotyped viruses. When Ty1 was expressed in mammalian cells as an Fc-fusion protein the potent neutralization could be further increased to ~12 ng/ml (Fig. 2a).

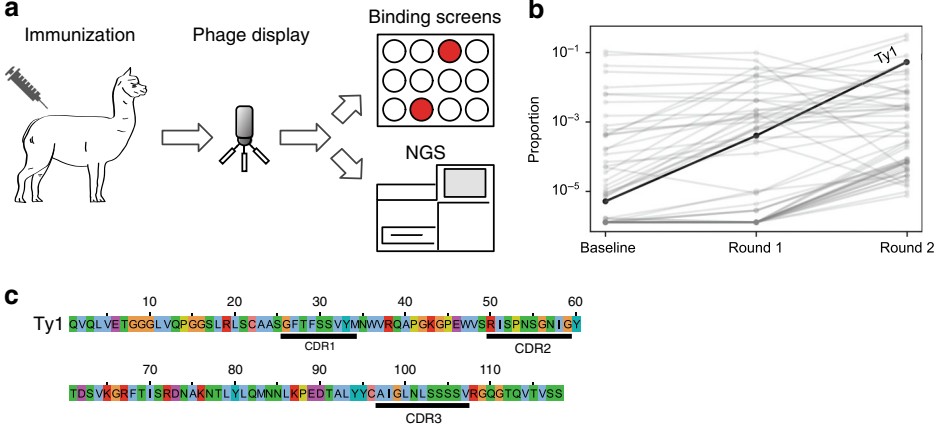

**Fig. 1 Nanobody discovery. a** Overview of the nanobody generation process. **b** Variant frequencies quantified by next generation sequencing (NGS) across successive enrichment steps. Identified using receptor-binding domain (RBD) bait, Ty1 exhibits the greatest total fold change of all nanobodies, increasing in proportion over 10,000-fold between initial and final libraries. **c** Sequence of Ty1. Complementarity-determining regions (CDRs) are indicated. Amino acid color labels; hydrophobic, blue; positive charge, red; negative charge, magenta; aromatic, cyan; polar, green; cysteine, pink; glycine, orange; proline, yellow.

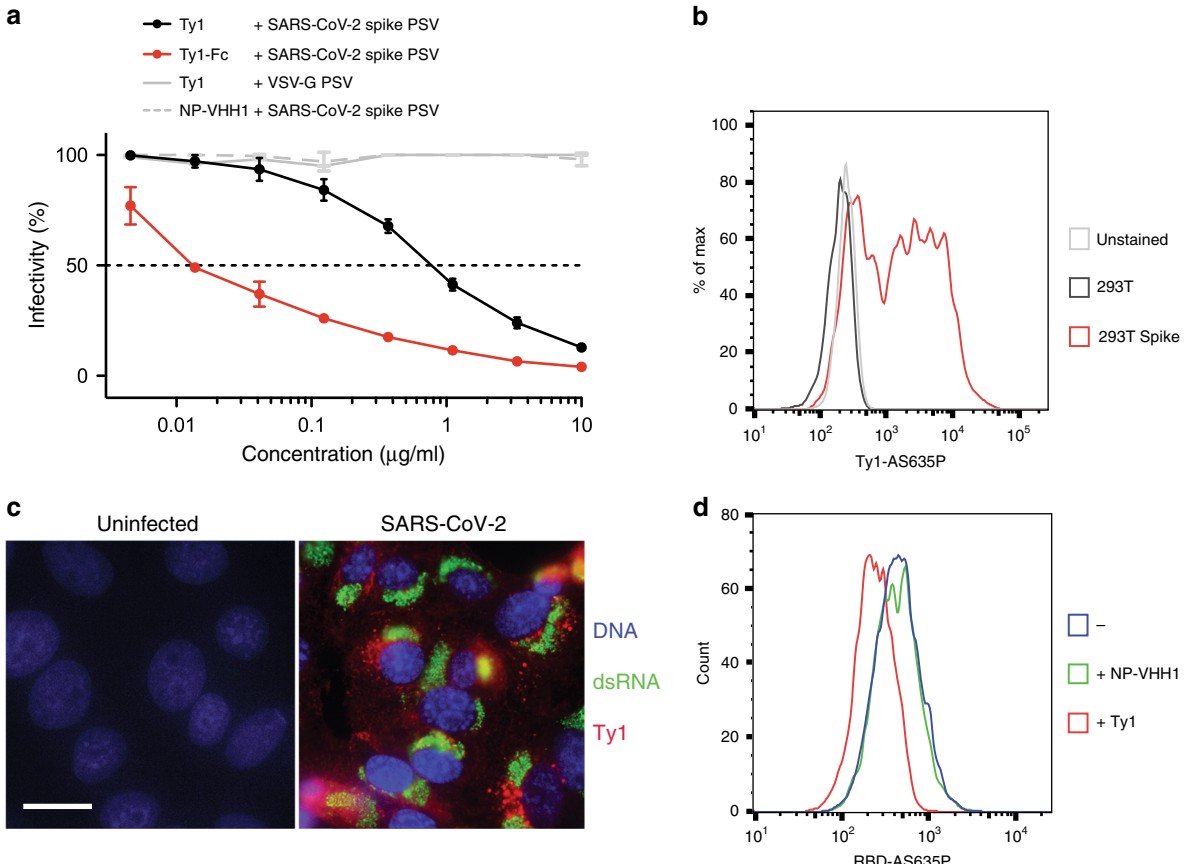

**Fig. 2 Ty1 neutralizes SARS-CoV-2 by binding to SARS-CoV-2 spike glycoprotein. a** VSV G or SARS-CoV-2 spike pseudotyped lentivirus (PSV) was incubated with a dilution series of Ty1, Ty1-Fc, or control nanobody (NP-VHH1[11]). Infectivity relative to cells infected with pseudotyped virus in the absence of nanobody is shown. Neutralization by Ty1 was repeated in duplicate across six assays, neutralization by Ty1-Fc was repeated in duplicate across two assays, and the error bars represent the standard deviation. **b** Cells were transfected with a plasmid harboring the SARS-CoV-2 spike for 24 h. Cells were fixed, permeabilized, and stained with Ty1-AS635P (black and red) or left unstained (gray). Cells were analyzed by flow cytometry. Cell counts are presented as % of max (representative histogram). **c** Vero E6 cells were infected with SARS-CoV-2 at a MOI of 1 for 24 h. Cells were fixed, permeabilized, and stained for DNA (blue), dsRNA (green), and with Ty1-AS635P (red). Pictures were taken by fluorescence microscopy and representative examples are shown. Scale bar, 20 μm. **d** ACE2 expressing HEK293T cells were trypsinized, fixed, and stained with RBD-AS635P alone (blue), or preincubated with NP-VHH1 (green) or Ty1 (red). Cells were analyzed by flow cytometry.

To confirm that Ty1 is directed specifically against the SARS-CoV-2 spike protein, we characterized the specificity of Ty1 by flow cytometry. We site-specifically conjugated a fluorophore to the C-terminus of Ty1 by means of a Sortase A reaction and copper-free click chemistry (Ty1-AS635P) and stained untransfected cells and cells transiently transfected with SARS-CoV-2 spike under permeabilizing conditions (Fig. 2b). While untransfected and unstained cells displayed similar signals, cells expressing the viral spike protein showed a strong shift in fluorescence intensity when stained with Ty1-AS635P. The apparent double peak likely reflected the varying efficiency of this transient transfection. To determine if the same probe can be exploited to recognize the viral spike protein in immunofluorescence, we infected Vero E6 cells with infectious SARS-CoV-2 at MOI 1 for 24 h, and stained the fixed and permeabilized cells with Ty1-AS635P and anti-dsRNA antibody (Fig. 2c). While uninfected cells showed no signal, infected cells were strongly labeled with both dsRNA antibody and Ty1-AS635P. Thus, Ty1 recognized the viral spike glycoprotein with high specificity in its native conformation in SARS-CoV-2-infected cells. Importantly, the low background in both experiments also suggested that Ty1 is a highly specific and suitable tool for research, diagnostics, and therapy.

To understand the mechanism of neutralization, we evaluated the effect of Ty1 on RBD binding to ACE2. We site-specifically conjugated a fluorophore to the C-terminus of the RBD (RBD-AS635P) and used this probe to stain ACE2 expressing HEK293T cells (Fig. 2d). Preincubation of RBD-AS635P with unlabeled Ty1 resulted in a strong reduction of ACE2 staining, while preincubation with the control nanobody NP-VHH1[11], specific for influenza A virus nucleoprotein NP, had no such effect. This result indicated that Ty1 directly prevents binding of SARS-CoV-2 RBD to its host cell receptor ACE2.

**Ty1 binds to RBD with high affinity.** Specific and high-affinity binding of Ty1 to the RBD was also demonstrated in kinetic bio layer interferometry (BLI) experiments. Dipping of surface-immobilized nanobodies into monomeric RBD solutions at 550 nM yielded binding responses with fast association kinetics and amplitudes reaching 1.5 nm only for Ty1 but not for NP-VHH1 (red and blue curves, respectively, in Fig. 3a). Titration experiments performed under normal (280 mM) and high salt (680 mM) conditions revealed concentration-dependent kinetic response curves for binding of RBD to Ty1 (Fig. 3b and Supplementary Fig. 1a, respectively). The derived semi-log concentration–response

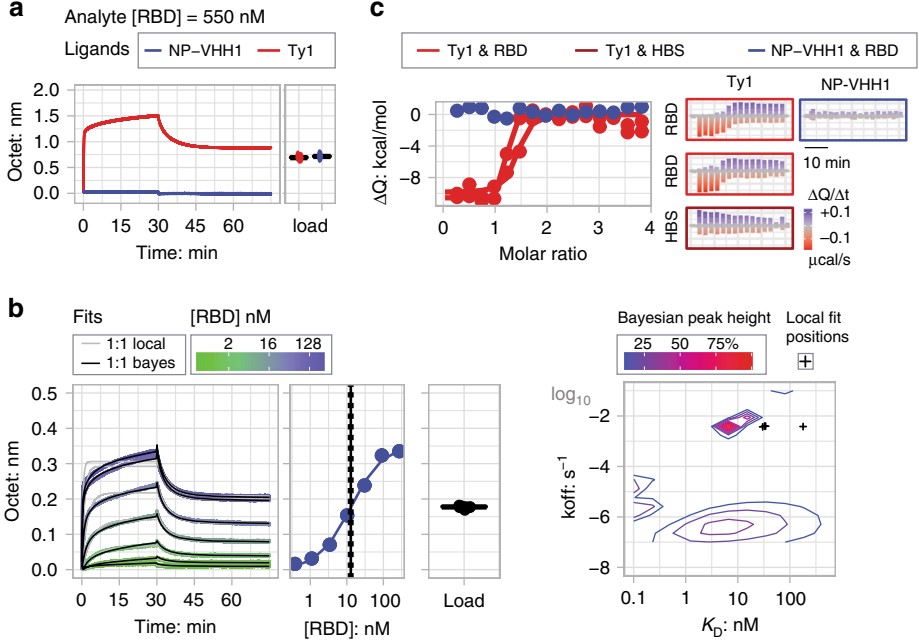

**Fig. 3 RBD binds to Ty1 with a $K_D$ of 5–10 nM. a** RBD bound to surface-immobilized Ty1 (red curves), but not to NP-VHH1 (blue curves). Almost equal nanobody immobilization levels of about 0.7 nm were obtained by first loading Ty1 and then NP-VHH1. In bio-layer interferometry (BLI), binding of molecules over time is recorded as sensorgrams recording the shift in wavelengths (unit: nm) due to an increase in the optical thickness of the surface layer. **b** (1st panel) RBD titration sensorgrams obtained at high salt concentrations revealed concentration-dependent responses. Sensorgrams are color-coded according to the log2 RBD concentration scale. Standard and Bayesian 1:1 binding models are shown as gray and black solid lines. (2nd panel) Pseudo-equilibrium response values plotted against the logarithmic RBD concentration revealed sigmoidal binding curves that were fit to the single-site interaction model[50] yielding $K_D$-values in the low nM range. $K_D$-values and standard deviations are shown as solid and dotted lines, respectively. (3rd panel) Sensor immobilization levels are shown as jittered box plots. (4th panel) Two-dimensional distribution of dissociation rate ($k_{off}$) and affinity ($K_D$) values obtained from the Bayesian and standard 1:1 model fits, visualized as densities according to depicted normalized height scale and single black crosses, respectively. See related Supplementary Fig. 1a. Plots for RBD/Ty1 titrations at normal salt condition with same legends and scales. **c** (1st panel) ITC demonstrated high-affinity binding of Ty1 to RBD with fitted $K_D$ and binding enthalpy ($\Delta H$) mean values of 9 nM (with estimated bounds of 1 and 70 nM) and −10 ± 0.5 kcal/mole (mean value ± standard deviation), respectively, for two binding experiments (red). Injection of NP-VHH1 into RBD yielded heat changes at background level (blue). (2nd panel) Baseline-corrected heat changes plotted for two Ty1/RBD, a single Ty1/HBS buffer, and NP-VHH1/RBD titration experiments. Negative and positive heat changes are colored according to the red-to-blue gradient. See related Supplementary Fig. 1b: Same figure on larger scale to highlight the Ty1 into buffer dilution spikes.

curves revealed sigmoidal line-shapes with fitted apparent $K_D$-values of 8 ± 1.5 and 13 ± 1.5 nm (mean value ± standard deviation) for binding at normal and high salt conditions, respectively. Local fits to individual sensorgrams applying the standard 1:1 binding model appeared reasonable for the association phases at lower to intermediate RBD concentrations, as well as for all dissociation curves when fits were allowed to stay above zero (gray lines Fig. 3b and S1A 1st panel). However, the model deviated from the observed data at higher RBD concentrations. Instead almost perfect fits were obtained when the same data were analyzed in terms of a Bayesian two-dimensional distribution of $K_D$ and $k_{off}$-rate constants to address heterogeneous ligand site populations on the sensor surface[12–14]. For the two titrations at normal and high salt conditions, distinct peaks at $K_D$ and $k_{off}$-rate values of 5–9 nM and $4–5 \times 10^{-3}$ s$^{-1}$ were obtained (Fig. 3b and Supplementary Fig. 1a 4th panel). In both conditions, a second elevated plateau with $K_D$ and $k_{off}$-rate values of about 7 nM and $1–4 \times 10^{-7}$ contributed significantly to the observed sensorgrams. Since most high-affinity protein–protein interactions in the nM-range have dissociation rates in the $1 \times 10^{-3}$ s$^{-1}$ range[15], we attribute the first defined peak as the relevant Ty1:RBD interaction. The second broad plateau is likely caused by RBD competition and rebinding effects on the sensor surface, as well as heterogeneous ligand populations[13,16,17]. The orthogonal biophysical method isothermal titration calorimetry (ITC)[18] confirmed the high affinity binding of Ty1 to RBD with a $K_D$ of 9 nM (with estimated

bounds of 1 and 70 nM) characterized by an exothermic enthalpy of about −10 ± 0.5 kcal/mole (Fig. 3c, left panel). Exothermic binding was already evident from the three initial relatively constant negative spikes that were caused by the injection of Ty1 to RBD (Fig. 3c, right panel). The amplitude of the following three to four spikes returned to baseline demonstrating saturation of the available RBD sites by Ty1 binding. Notably, return to baseline was accompanied by the appearance of preceding positive spikes (Fig. 3c, left panel and Supplementary Fig. 1b). These spikes were also detected when Ty1 was injected into the buffer (HBS) and thus treated as Ty1 dilution effects during data analysis. Injection of NP-VHH1 into RBD did not cause any binding or dilution heat changes above background noise. It should be noted that the ITC measurements were performed at the lowest possible protein concentrations to derive $K_D$-values in the low nM range, while still being able to detect interaction heat above background noise signals that were at about −0.17 μcal/s (maximum spike amplitude) and ±0.05 μcal/s, respectively. Altogether, we concluded from these results that RBD bound to Ty1 with high affinity of about 5–10 nM.

**Ty1 binds to the RBD in 'up' and 'down' conformation.** To understand the structural basis underlying the potent neutralization of SARS-CoV-2 we performed a cryo-EM structure determination of the prefusion-stabilized spike ectodomain in

complex with Ty1. The cryo-EM reconstruction reaches an overall resolution of 2.9 Å (0.143 FSC; Supplementary Table 1) with strong variation of estimated local resolution from high resolution in the core of the spike trimer to relatively low resolution in the top of the spike (Fig. 4a and Supplementary Fig. 2). Nevertheless, this reconstruction clearly shows that the spike retains only one main conformation with one RBD 'up' and two RBDs 'down'. Importantly, all three RBDs are decorated in their upper parts with a Ty1 nanobody. The nanobodies retain a similar binding orientation to the RBD whether the RBD is found in the 'up' or 'down' conformation (Fig. 4a, b) and each has a solvent-excluded surface area of ~860 Å$^2$, which is in line with the strong affinity observed in the biophysical-interaction studies. Primary interactions with the RBD are through the CDRs. Specifically, CDR1 interacts primarily with RBD T470 and V483-E484, and CDR3 interacts primarily with RBD Y449, F490, and Q493. Interestingly, CDR2 does not form any major interactions with the RBD, instead it stabilizes the conformation of CDR1 in the RBD bound mode and thereby acts indirectly to potentiate the Ty1–RBD binding.

Since ACE2 can only be bound by an RBD in the 'up' conformation, the cryo-EM reconstruction clearly shows that ACE2 binding is sterically hindered from two sides (Fig. 4c). Specifically, ACE2 binding is blocked both by the Ty1 nanobody bound to the RBD in the 'up' conformation and the neighboring RBD in the 'down' conformation. Hence, ACE2 binding is sufficiently hindered with any two of the available three binding RBD sites in the spike trimer.

## Discussion

The current coronavirus pandemic has drastic consequences for the world's population, and vaccines, antibodies, or antivirals are urgently needed. Neutralizing antibodies can block virus entry at an early step of infection and potentially protect individuals that are at high risk of developing severe disease. We report the identification and characterization of a SARS-CoV-2 RBD-specific single domain antibody fragment (nanobody) termed Ty1 that potently neutralizes the virus. We identified Ty1 by binding assay after two consecutive rounds of phage display, simultaneously monitoring sequence enrichment by NGS. Although Ty1 exhibited the greatest fold-enrichment in the NGS analysis, multiple additional nanobodies exhibited enrichment of varying extent across both rounds. As the correlation between phage display enrichment and neutralization is likely imperfect, further analyses of our libraries may yield other potent SARS-CoV-2 neutralizing nanobodies. In addition to neutralization activity, we also show that Ty1 can be used as a detection reagent in flow cytometry and immunofluorescence demonstrating its suitability as a research tool and for diagnostics. Glycans on spike glycosylation sites N165, N234, and N343 shield the RBD from antibodies, especially when the RBD is in down conformation[19,20]. Indeed, in the RBD-down conformation, the glycan on N165 points towards the Ty1-binding epitope, likely not leaving sufficient space to accommodate a conventional antibody. In agreement with that, Fab fragments from convalescent patients bound the RBD only in the up conformation and to an epitope that only minimally overlaps with the Ty1 epitope[21].

It should be noted that the nanobody Ty1 can be readily produced in bacteria at very high yield (in excess of 30 mg/L culture), making it an excellent candidate for a low-cost, scalable antiviral agent against SARS-CoV-2, and we provide the amino acid sequence, encouraging direct exploitation as such. Interestingly, while Ty1 contains the hallmark (hydrophobic) amino acids of variable-heavy chains in framework 2, only one arginine

(instead of tryptophan) in framework 4 demonstrates that this antibody fragment derives from a heavy-chain only antibody[22]. Nevertheless, Ty1 expresses extremely well, but exchanging the hydrophobic residues in framework 2 may further improve this nanobody. While nanobodies capable of binding SARS-CoV-2 spike have recently been isolated, these were generated after SARS-CoV-1 spike immunization[23], or PCR maturation[24]. Also, in both cases a fusion to human Fc domain is required for neutralization of SARS-CoV-2, precluding expression in bacterial culture. Naive libraries of human single-domain antibodies (sdAbs) have also been screened to identify SARS-CoV-2 spike-specific nanobodies[25,26], but they lack detailed structural information. Other synthetic RBD-specific nanobodies have been published, but they lack information on their neutralization potential[27]. Ty1 represents the first single-domain antibody isolated from an animal specifically immunized with a SARS-CoV-2 protein.

Future work will aim to improve the potency and potential efficacy of Ty1 through various strategies. For example, mutational scanning may yield potency improvements to Ty1. Also, since Ty1 already neutralizes as a monomeric protein, the generation of homodimeric or trimeric fusion constructs is expected to further increase its neutralization activity. Indeed, fusion of Ty1 to a human IgG1-Fc dramatically improved the IC$_{50}$ of this molecule, to ~12 ng/ml. Additional strategies will explore linker-based constructs that chain multiple copies of Ty1 together, which may provide similar improvements in potency while retaining the possibility of being expressed in bacteria. Ty1 may additionally be a useful component of a bi-specific or tri-specific antibody, which could combine epitope specificities to increase the mutational barrier to viral escape. Based on our work, we hope that Ty1 will be investigated as a candidate for antiviral therapy.

## Methods

**Cells and virus.** Vero E6 cells (ATCC-CRL-1586) and HEK293T cells (ATCC-CRL-3216) were maintained in Dulbecco's modified Eagle medium (Gibco) supplemented with 10% fetal calf serum and 1% penicillin–streptomycin and cultured at 37 °C in a humidified incubator with 5% CO$_2$. A HEK293T cell line engineered to overexpress human ACE2 (HEK293T-ACE2) was generated by the lentiviral transduction of HEK293T cells. Briefly, lentiviruses were produced by co-transfecting HEK293T cells with a plasmid encoding VSV-G (Addgene cat#12259), a lentiviral Gag-Pol packaging plasmid (Addgene cat#8455), and a human ACE2 transfer plasmid. Virions were harvested from the supernatant, filtered through 0.45 μm filters, and used to transduce HEK293T cells. All cell lines used for experiments were negative for *Mycoplasma* as determined by PCR. Infectious SARS-CoV-2[28] was propagated in Vero E6 cells and titrated by plaque assay.

**Proteins and probes.** The plasmid for expression of the SARS-CoV-2 prefusion-stabilized spike ectodomain with a C-terminal T4 fibritin trimerization motif was obtained from ref. [2]. The plasmid was used to transiently transfect FreeStyle 293F cells using FreeStyle MAX reagent (Thermo Fisher Scientific). The S ectodomain was purified from filtered supernatant on Streptactin XT resin (IBA Lifesciences), followed by size-exclusion chromatography on a Superdex 200 in 5 mM Tris pH 8, 200 mM NaCl.

The RBD domain (RVQ-VNF) of SARS-CoV-2 was cloned upstream of an enterokinase cleavage site and a human IgG1 Fc. This plasmid was used to transiently transfect FreeStyle 293F cells using the FreeStyle MAX reagent. The RBD-Fc fusion was purified from filtered supernatant on Protein G Sepharose (GE Healthcare). The protein was cleaved using bovine enterokinase (GenScript) leaving a FLAG-tag at the C-terminus of the RBD. Enzyme and Fc-portion were removed on HIS-Pur Ni-NTA resin (Thermo Fisher Scientific) and Protein G sepharose (GE Healthcare), respectively, and the RBD was purified by size-exclusion chromatography on a Superdex 200 in 50 mM Tris pH 8, 200 mM NaCl.

In addition, the RBD domain (RVQ-VNF) was cloned upstream of a Sortase A recognition site (LPETG) and a 6xHIS tag and expressed in FreeStyle 293F cells as described above. RBD-HIS was purified from filtered supernatant on His-Pur Ni-NTA resin, followed by size-exclusion chromatography on a Superdex 200.

The nanobodies were cloned for expression in the pHEN plasmid with a C-terminal Sortase recognition site (LPETG) and a 6xHIS tag. This plasmid was used to transform BL21 cells for periplasmic expression. Expression was induced with 1 mM IPTG at OD600 = 0.6; cells were grown overnight at 30 °C.

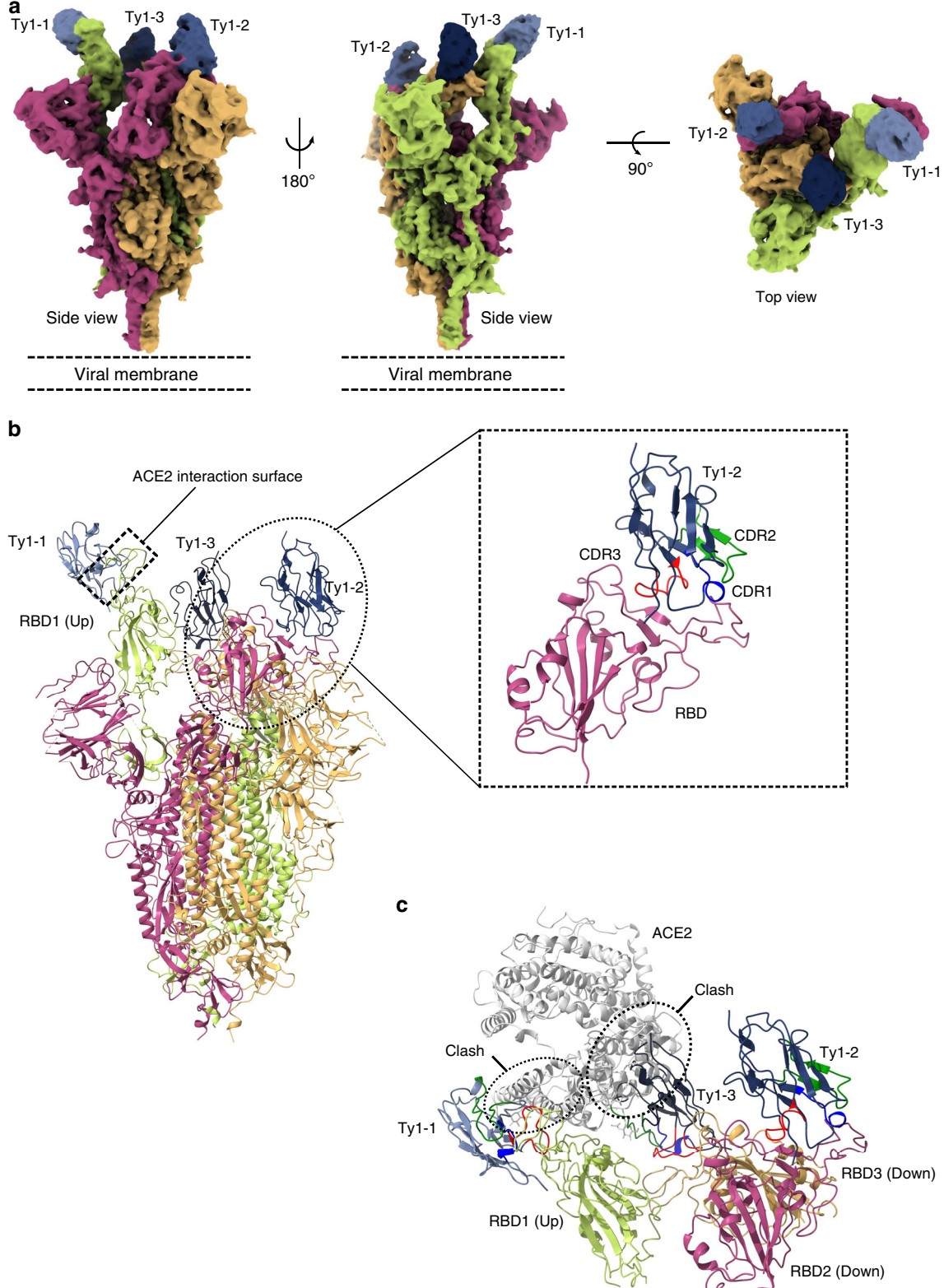

**Fig. 4 Ty1 binds to the RBD in 'up' and 'down' conformation and prevents ACE2 engagement. a** Cryo-EM reconstruction to an overall resolution of 2.9 Å (0.143 FSC) of the spike trimer with three bound molecules of Ty1. **b** Atomic model (cartoon representation) of trimeric spike in complex with three molecules of Ty1. Three chains of spike are shown in three different colors. The RBD of chain A (light green) is present in 'up' conformation while the other two RBDs are captured in 'down' conformation. The ACE2 interaction surface of RBD1 and the Ty1 interaction surface is highlighted. Magnified view of RBD2 (in 'down' conformation) and Ty1 interaction is shown. CDR1, 2, and 3 of Ty1 are shown in blue, green, and red, respectively. **c** Ty1 shows a two-pronged inhibition of ACE2 receptor binding through binding the RBD in the 'up' conformation and by binding to the neighboring RBD in the 'down' conformation. Binding of Ty1 to RBDs (both 'up' and 'down') would make ACE2 interaction surface inaccessible for ACE2.

Nanobodies were retrieved from the periplasm by osmotic shock and purified by Ni-NTA affinity purification and size-exclusion chromatography.

Biotinylated and fluorescent probes were generated using Sortase A as described in refs. [29,30]. In brief, nanobodies were site-specifically biotinylated on the C-terminus using Sortase A 5 M. Nanobody at a concentration of 50 μM was incubated with sortase A 5 M (5 μM), GGGK-biotin (200 μM) in 50 mM Tris, pH 7.5, 150 mM NaCl, 10 mM CaCl₂, for 2 h at 25 °C. Unreacted nanobody and sortase was removed with Ni-NTA resin and excess GGGK-biotin was removed using Zeba spin desalting columns (0.5 ml, 7k MWCO, Thermo Fisher Scientific).

To generate the fluorescently labeled probes, first a dibenzocyclooctyne-amine (DBCO-amine, Sigma Aldrich) was attached via sortase A to the nanobody or the RBD (reaction conditions: 50 μM RBD or nanobody, 50 μM Sortase A 5 M, 8 mM DBCO-amine in 50 mM Tris pH 7.5, 150 mM NaCl, 10 mM CaCl₂, 2 h, 25 °C). Unreacted probe, sortase and excess DBCO-amine were removed using Ni-NTA resin and PD-10 columns (GE Healthcare), respectively. Abberior Star 635P-azide (Abberior GMBH) was attached to the DBCO-labeled proteins in a copper-free click chemistry reaction. Unreacted fluorophore was removed on PD-10 column (RBD) or size-exclusion chromatography (nanobody).

For mammalian expression, the sequence encoding the nanobody Ty1 was cloned upstream of a human IgG1. This plasmid was used to transiently transfect FreeStyle 293F cells using the FreeStyle MAX reagent. The Ty1-Fc fusion was purified from filtered supernatant on Protein G Sepharose followed by size-exclusion chromatography.

**Alpaca immunization.** Alpaca immunization and phage display was performed similarly as described in refs. [31,32]. In brief, the adult male alpaca Tyson at Pre-Clinics, Germany, was immunized four times in a 60-day immunization schedule. SARS-CoV-2 S1-sheep-Fc (Native Antigen Company, SKU: REC31806) was used for the first two immunizations, and SARS-CoV-2 RBD produced in FreeStyle 293F cells was used for the last two immunizations. The animal study protocol was approved by the PreClinics animal welfare officer commissioner and registered under the registration No. 33.19-42502-05-17A210 at the Lower Saxony State Office for Consumer Protection and Food Safety—LAVES and is compliant with the Directive 2010/63/EU on animal welfare.

**Library generation and nanobody isolation.** Four days after the final boost, RNA was isolated from PBMCs (RNA Plus mini kit, Qiagen). For cDNA synthesis, SuperScript III RT (Thermo Fisher Scientific) was used with a combination of oligo (dT), random hexamers, or gene-specific primers (AL.CH2, ATGGAGAGGAC GTCCTTGGGT, and AL.CH2.2 TTCGGGGGGAAGAYRAAGAC)[32]. All primer sequences are listed in Supplementary Table 2. Nanobody sequences were PCR amplified and cloned into a phagemid vector for expression as pIII fusion. TG1 cells (Lucigen) were transformed with this library by electroporation.

Cells were inoculated with VCSM13 helper phage, and the resulting phage was enriched in two consecutive rounds of phage display on RBD immobilized on magnetic beads. After the second round of phage display, individual bacterial colonies were picked in a 96-well format, grown until OD = 0.6 and nanobody expression was induced by addition of 1 mM IPTG. After 16 h incubation at 30 °C, bacterial supernatant was used as primary detection reagent in an ELISA coated with RBD or S ectodomain. Bound nanobodies were detected with anti-E tag (Bethyl laboratories, 1:10,000) secondary antibody. Positive clones were sequenced and cloned into the pHEN expression vector for further characterization.

**Amino acid sequence of Ty1.** QVQLVETGGGLVQPGGSLRLSCAASGFTFSS VYMNWVRQAPGKGPEWVSRISPNSGNIGYTDSVKGRFTISRDNAKN TLYLQMNNLKPEDTALYYCAIGLNLSSSSVRGQGTQVTVSS

**NGS and analysis of nanobody libraries.** Plasmids from nanobody libraries before enrichment, and after each enrichment step, were amplified for 13 cycles using Q5 High-Fidelity 2X Master Mix (NEB) according to manufacturer's instructions, using primers: NB-NGS-FW: CACTCTTTCCCTACACGACGCTC TTCCGATCTCTCGCGGGCCCAGCCGGCCATGG and NB-NGS-RV: GGAGTTC AGACGTGTGCTCTTCCGATCTACCGGCGCACCACTAGTGCA, annealing at 72 °C. Illumina indexing primers were added using an additional 9 cycles, with Kapa HiFi. Amplicons were size selected using Agencourt AMPure XP beads (bead ratio: 1:1), and were pooled at ratios of 6:2:1 for pre:post-1:post-2 libraries, to account for the reduction in diversity expected during enrichment, and sequenced on an Illumina MiSeq using the MiSeq Reagent Kit v3 (2 × 300) MS-102-3003.

Paired-end reads were merged using USEARCH11[33], and then processed in the Julia language, primarily using the NextGenSeqUtils.jl package[34] (analysis code is available here: https://github.com/MurrellGroup/Ty1). Briefly, reads are trimmed of primer sequences, and deduplicated, maintaining read frequencies. Variant frequencies are calculated as combined frequency of any reads matching a variant within 3% nucleotide divergence, using a kmer-based distance approximation for rapid database search. Any reads with counts >3 from the second enrichment library are searched for their variant frequencies across all databases. When calculating enrichment, to avoid zeros due to sampling and to regularize against over-sensitivity to low-frequency baseline variants, all frequencies are increased by the reciprocal of the size of the pre-enrichment database.

**Neutralization assay.** Pseudotyped viruses were generated by the co-transfection of HEK293T cells with plasmids encoding the SARS-CoV-2 spike protein harboring an 18 amino acid truncation of the cytoplasmic tail[35], a plasmid encoding firefly luciferase, and a lentiviral packaging plasmid (Addgene cat#8455) using Lipofectamine 3000 (Invitrogen). Media was changed 12–16 h after transfection, and pseudotyped viruses were harvested at 48 and 72 h post transfection, filtered through a 0.45 μm filter, and stored at −80 °C until use. Pseudotyped virus neutralization assays were adapted from protocols previously validated to characterize the neutralization of HIV[36], but with the use of HEK293T-ACE2 cells. Briefly, pseudotyped viruses sufficient to generate ~100,000 RLUs were incubated with serial dilutions of nanobodies for 60 min at 37 °C. Approximately 15,000 HEK293T-ACE2 cells were then added to each well and the plates were incubated at 37 °C for 48 h. Luminescence was then measured using Bright-Glo (Promega) per the manufacturer's instructions on a GM-2000 luminometer (Promega) with an integration time of 0.3 s.

**Flow cytometry.** Cells were trypsinized and fixed in 4% formaldehyde/PBS and stained with RBD-AS635P under non-permeabilizing conditions or with Ty1-AS635P under permeabilizing conditions. Fluorescence was quantified using a BD FACSCelesta and the FlowJo software package.

**Immunofluorescence.** Vero E6 cells were seeded onto coverslips in a 24-well plate and incubated overnight at 37 °C/5% CO₂. Cells were infected with SARS-CoV-2 at a MOI of 1 for 24 h. Cells were fixed with 4% (v/v) formaldehyde, permeabilized in 0.5% Triton X-100 and blocked in 5% horse serum. Cells were incubated with anti-dsRNA antibody (1:2000, J2 Scicons, RNT-SCI-10010200) for 1 h at room temperature followed by 1 h staining with the secondary antibody anti-mouse-Alexa Fluor 488 (1:2000, Thermo Fisher Scientific, A-21202), Hoechst (1:1000, Invitrogen) and Ty1-AS635P (0.5 μg/ml). Coverslips were mounted in mounting media and images were obtained using Zeiss Axiovert microscope and processed using Adobe Photoshop.

**Biophysical BLI and ITC.** BLI was performed using single-use high-precision streptavidin biosensors (SAX) on an eight-channel Octet RED instrument according to manufacturer's protocols (Fortebio)[16]. Assays were performed in 2xPBS comprising 0.05% Tween-20 (PBST). Biotinylated nanobodies Ty1 and NP-VHH1 were loaded at concentrations between 30 and 250 nM followed by quenching using biocytin to reach final sensor loads of between 0.15 and 0.7 nm. For the comparative binding test, the eight sensors were divided into two sets, each comprising double sample as well as single reference and single control sensors. Sample and reference sensors were loaded with respective nanobodies. The SAX control was only quenched. Loading of the two sets was performed consecutively to reach similar immobilization levels, while subsequent association and dissociation phases were performed simultaneously. For association, the sample and control sensors were dipped into RBD, while the reference sensor was dipped into PBST. For titration experiments, all sensors were loaded simultaneously. During association one of the sensors was used as reference and only dipped into PBST. Raw data were preprocessed, analyzed, and fitted by applying the 1:1 binding model as implemented in the manufacturer's software. Bayesian analysis to obtain the two-dimensional distribution of $K_D$ and $k_{off}$-rate values were performed using Evilfit[12–14]. The shown titration data were processed applying reference sensor subtraction and Savitzky–Golay filter operations.

For ITC, proteins were exchanged to 2xHBS-buffer (50 mM HEPES, 300 mM NaCl, pH 7.5) and isolated as single peak populations by Superdex-200 HR10/300 size-exclusion chromatography. ITC measurements were performed using an ITC200 calorimeter (GE Healthcare). The cell temperature was set to 37 °C and the syringe stirring speed to 750 rpm. Before each experiment, the RBD and nanobodies were loaded into the cell and syringe at concentrations of 4 and 75 μM, respectively. Data and binding parameters were analyzed using the MicroCal PeakITC software (Malvern). The integrated heat versus molar ratio plots of the Ty1:RBD interactions were obtained by subtracting the Ty1 dilution heat uptake from the binding data. The NP-VHH1:RBD data were only baseline-corrected, since dilution effects were not evident.

Raw and processed BLI/ITC data were imported into Rstudio for visualization and further analysis[37–39]. Data along with analysis R scripts will be made publicly available via Github and/or DataDryad.

**Cryo-EM sample preparation and imaging.** Spike trimer (0.7 mg/ml) and Ty1 (1.3 mg/ml) were mixed in a 1:8 molar ratio and incubated on ice for 5 min. A 3-μl aliquot of the sample solution was applied to glow-discharged CryoMatrix holey grids with amorphous alloy film (Zhenjiang Lehua Technology) in a Vitrobot Mk IV (Thermo Fisher Scientific) at 4 °C and 100% humidity (blot 10 s, blot force 3).

Cryo-EM data collection was performed with EPU 2.7 (Thermo Fisher Scientific) using a Krios G3i transmission-electron microscope (Thermo Fisher Scientific) operated at 300 keV in the Karolinska Institutet 3D-EM facility. Images were acquired in nanoprobe EFTEM mode with a slit width of 10 eV using a GIF 967 energy filter (Ametek) and a K3 detector (Ametek) during 2.4 s with a dose rate of 4.1 e⁻/px/s resulting in a total dose of 38 e⁻/Å² fractionated into 40 movie frames. Motion correction, CTF-estimation, Fourier binning (to 1.02 Å/px),

picking and extraction in 428 pixel boxes were performed on the fly using Warp[40]. A total of 13,589 micrographs were selected based on an estimated resolution cutoff of 4 Å and defocus below 2 microns and 573,036 particles were picked by Warp. Extracted particles were imported into cryoSPARC v2.15.0[41] for 2D classification, 3D classification, and non-uniform 3D refinement. The particles were processed with C1 symmetry throughout. After 2D classification (300 classes) 354,678 particles were retained and used to build three ab-initio 3D reconstructions. These were further processed for heterogeneous refinement that resulted in one reconstruction showing high-resolution structural features in the core of the spike. One round of homogenous refinement followed by non-uniform refinement resulted in a final reconstruction to an overall resolution of 2.9 Å (0.143 FSC) using 210,832 particles. Localized reconstruction[42] were performed using particles where all parts of the spike except the N-terminal domains, the RBDs, and the nanobodies had been subtracted[43]. The combined effects of these two approaches significantly increased the level of density detail in the upper part of the spike.

**Model building and structure refinement**. A structure of the 2019-nCoV spike protein trimer[2] (PDB: 6VSB) was used as a starting model for model building. The model was extended and manually adjusted in COOT[44]. The nanobody structure was homology modeled using SWISS-MODEL[45] taking PDB:5JMR[46] as a template. The missing regions of the RBD domains were built based on the RBD-ACE2 crystal structure (PDB: 6LZG)[47]. For model building and refinements, a composite map was made using PHENIX[48] utilizing the particle center-of-mass focused reconstruction and the map from the localized reconstruction described above.

Structure refinement and manual model building were performed using COOT and PHENIX in interspersed cycles with secondary structure and geometry restrained. All structure figures and all EM density-map figures were generated with UCSF ChimeraX[49].

**Reporting summary**. Further information on research design is available in the Nature Research Reporting Summary linked to this article.

## Data availability

The sequence of Ty1 is deposited in the NCBI GenBank sequence data base and is available under the accession code MT784731. BLI and ITC data are available in https://github.com/derpaule/Ty1_octet_itc and https://doi.org/10.5061/dryad.gb5mkkwmz, respectively. Next generation sequencing data is deposited at the SRA, under BioProject ID PRJNA638614. Jupyter notebooks to reproduce the NGS data processing are available at: https://github.com/MurrellGroup/Ty1. The cryo-EM density map of SARS-CoV-2 spike glycoprotein with Ty1 nanobodies bound was deposited in the Electron Microscopy Data Bank (EMDB) with accession code EMD-11526. The corresponding model was deposited in the Protein Data Bank (PDB) with accession code 6ZXN. Source data are provided with this paper.

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

## Acknowledgements

SARS-CoV-2 virus was received from Jonas Klingström, KI. The authors thank James Voss and Deli Huang for providing reagents. We thank Jason McLellan for providing the expression plasmid of the stabilized SARS-CoV-2 spike protein. We thank PreClinics for their fast and efficient services as well as Gunnar Schulte (KI), Per-Åke Nygren (KTH Stockholm), and Tomas Nyman (Protein Science Facility at KI/SciLifeLabs) for providing access to BLI and ITC instruments. All cryo-EM data were collected in the Karolinska Institutet 3D-EM facility. We also thank Fondation Dormeur, Vaduz for generous support of flow cytometry equipment. This project has received funding from the European Union's Horizon 2020 research and innovation program under grant agreement No. 101003653 (CoroNAb), to G.M.M., G.B.K.H., and B.M. and from project grants from the Swedish Research Council to B.M. (2018-02381), B.M.H. (2017-6702 and 2018-3808) and to G.M.M. (2018-03914 and 2018-03843).

## Author contributions

L.H., L.V.P., D.J.S., T.S., A.M.-M., M.C. and B.M. performed experiments and analyzed data. H.D. and B.M.H. collected and processed cryo-EM data, built the model, and performed the structural interpretation of the data with the corresponding paper sections. A.A. and G.B.K.H. gave critical advice. L.H., B.M. and G.M.M. conceived the study and wrote the initial draft. All authors contributed to the final draft.

## Funding

## Competing interests

The authors declare no competing interests.
