## [Peer Review File · Nature Communications]

Reviewers' Comments:

Reviewer #1:

Remarks to the Author:

Hanke et al. present a fine study on nanobody (Ty1) isolated from the alpaca Tyson against the SARS-CoV2 spike protein specifically the RBD of the spike (immunisations with spike trimer and later only RBD). Using a pseudotyped lentivirus model expressing SARS-CoV2 spike protein for infection of HEK293 expressing ACE2 they find that Ty1 blocks infection. Lentivirus expressing VSV-G is not functional and a nanobody against influenza nucleoprotein A shows no effect. They characterise the binding by BLI and ITC to find consistently a 5-10 nM binding affinity, and they characterise the complex using cryo-EM determining the structure at overall 2.9 Å resolution. These are fine and consistent results, and they present a good report to also highlight the implications and requirements for clinical application of their findings.

It would be nice to compare to recent results for glycosylation dynamics/shielding of the spike protein (see e.g. <https://www.biorxiv.org/content/10.1101/2020.05.20.103325v1> and <https://www.ncbi.nlm.nih.gov/pmc/articles/PMC7217288/>). Furthermore, to recent studies of human antibody binding sites [https://www.cell.com/cell/pdf/S0092-8674\(20\)30757-1.pdf](https://www.cell.com/cell/pdf/S0092-8674(20)30757-1.pdf) - this could go to the discussion section in a minor revision.

minor point

The front page states an equal contribution note, but I don't see it applied to anyone?

Reviewer #2:

Remarks to the Author:

The consortium identified an affinity-matured nanobody against RBD of SARS-Cov2, retrieved from an immune alpaca Nb library. The selection was performed after only two rounds of phage display panning, and enrichments were checked by NGS. The highest enriched clone, named Ty1 was further analysed in a virus neutralisation capacity, in staining S protein generating cells. The affinity of binding was measured by bio interferometry and ITC. Cryo-EM was employed to identify the binding site. Interestingly, this nanobody binds simultaneously to the ACE2 accessible and in the two non-accessible conformers.

This reviewer is impressed by the quality and quantity of work that has been achieved in a short period of time. The experiments are well performed and the resulting data are highly interesting and important. Nevertheless, I would like to suggest a few minor amendments that could be taken into account.

Line 99 (and also later) the authors are using NP-VHH1 as a non-RBD binding control. However, I missed the specificity of this nanobody.

Line 202 and later, human FC is probably human Fc (from IgG1?)

Line 250: the sequence of the Ty1 nanobody should start with QVQL, the MA is presumably

a leftover of the pel B leader signal.

It is surprising that it is nowhere mentioned that the sequence of the VHH in FR2 corresponds to the VH germline sequences. V at pos 37 (Kabat) and PEW at pos 45-47. However, the R instead of W at pos 103 proves that it is a VHH (part of a heavy chain antibody) as explained in Muyldermans (2013) *Ann Rev Biochem.* 82.

It is also surprising that the authors missed the paper by Justin D Walter (Market Seeger lab) submitted to BioRxiv doi.org/10.1101/2020.04.16.045419; while they refer to the paper by Chi, X in BioRxiv ref 21) This paper Walter et al. should be added.

Reviewer #1 (Remarks to the Author):

Hanke et al. present a fine study on nanobody (Ty1) isolated from the alpaca Tyson against the SARS-CoV2 spike protein specifically the RBD of the spike (immunisations with spike trimer and later only RBD). Using a pseudotyped lentivirus model expressing SARS-CoV2 spike protein for infection of HEK293 expressing ACE2 they find that Ty1 blocks infection. Lentivirus expressing VSV-G is not functional and a nanobody against influenza nucleoprotein A shows no effect. They characterise the binding by BLI and ITC to find consistently a 5-10 nM binding affinity, and they characterise the complex using cryo-EM determining the structure at overall 2.9 Å resolution. These are fine and consistent results, and they present a good report to also highlight the implications and requirements for clinical application of their findings.

We thank the reviewer for the very positive review.

It would be nice to compare to recent results for glycosylation dynamics/shielding of the spike protein (see e.g. <https://www.biorxiv.org/content/10.1101/2020.05.20.103325v1> and <https://www.ncbi.nlm.nih.gov/pmc/articles/PMC7217288/>). Furthermore, to recent studies of human antibody binding sites [https://www.cell.com/cell/pdf/S0092-8674\(20\)30757-1.pdf](https://www.cell.com/cell/pdf/S0092-8674(20)30757-1.pdf) - this could go to the discussion section in a minor revision.

We thank the reviewer for this suggestion. This is indeed interesting and we have added a discussion of this in the Discussion on lines 198-204 in the revised manuscript.

minor point

The front page states an equal contribution note, but I don't see it applied to anyone? McInerney and Murrell contributed equally. This is now fixed.

Reviewer #2 (Remarks to the Author):

The consortium identified an affinity-matured nanobody against RBD of SARS-Cov2, retrieved from an immune alpaca Nb library. The selection was performed after only two rounds of phage display panning, and enrichments were checked by NGS. The highest enriched clone, named Ty1 was further analysed in a virus neutralisation capacity, in staining S protein generating cells. The affinity of binding was measured by bio interferometry and ITC. Cryo-EM was employed to identify the binding site. Interestingly, this nanobody binds simultaneously to the ACE2 accessible and in the two non-accessible conformers.

This reviewer is impressed by the quality and quantity of work that has been achieved in a short period of time. The experiments are well performed and the resulting data are highly interesting and important. Nevertheless, I would like to suggest a few minor amendments that could be taken into account.

We thank the reviewer for this positive review of our work and for the suggestions, which we have addressed below and in the revised manuscript.

Line 99 (and also later) the authors are using NP-VHH1 as a non-RBD binding control. However, I missed the specificity of this nanobody.

We have now included the specificity of this nanobody (influenza A virus nucleoprotein, NP; reference 11) in the main text on lines 125-126 in the revised manuscript.

Line 202 and later, human FC is probably human Fc (from IgG1?)

We have corrected this error.

Line 250: the sequence of the Ty1 nanobody should start with QVQL, the MA is presumably a leftover of the pel B leader signal.

We have now fixed this error in the text on line 322 of the revised manuscript and in Figure 1c.

It is surprising that it is nowhere mentioned that the sequence of the VHH in FR2 corresponds to the VH germline sequences. V at pos 37 (Kabat) and PEW at pos 45-47. However, the R instead of W at pos 103 proves that it is a VHH (part of a heavy chain antibody) as explained in Muyldermans (2013) Ann Rev Biochem. 82.

We thank the reviewer for this suggestion. We now include a short discussion of this in the Discussion on lines 215-217 of the revised manuscript.

It is also surprising that the authors missed the paper by Justin D Walter (Market Seeger lab) submitted to BioRxiv doi.org/10.1101/2020.04.16.045419; while they refer to the paper by Chi, X in BioRxiv ref 21) This paper Walter et al. should be added.

We agree with this suggestion and now cited the preprint from Walter et al. as reference 27 of the revised manuscript.